

# Stable isotope compositions of precipitation over Central Asia

Junqiang Yao[1], Xinchun Liu[1] and Wenfeng Hu[2]

[1] Institute of Desert Meteorology, China Meteorological Administration, Urumqi, Xinjiang, China
[2] Fuyang Normal University, Fuyang, Xinjiang, China

## ABSTRACT

Central Asia is one of the driest regions in the world with a unique water cycle and a complex moisture transport process. However, there is little information on the precipitation $\delta^{18}O$ content in Central Asia. We compiled a precipitation $\delta^{18}O$ database from 47 meteorological stations across Central Asia to reveal its spatial-temporal characteristics. We determined the relationship between precipitation $\delta^{18}O$ and environmental variables and investigated the relationship between $\delta^{18}O$ and large-scale atmospheric circulation. The Central Asia meteoric water line was established as $\delta^2H = 7.30 \, \delta^{18}O + 3.12$ ($R^2 = 0.95$, $n = 727$, $p < 0.01$), and precipitation $\delta^{18}O$ ranged from +2‰ to −25.4‰ with a mean of −8.7‰. The precipitation $\delta^{18}O$ over Central Asia was related to environmental variables. The $\delta^{18}O$ had a significant positive correlation with temperature, and the $\delta^{18}O$-temperature gradient ranged from 0.28‰/°C to 0.68‰/°C. However, the dependence of $\delta^{18}O$ on precipitation was unclear; a significant precipitation effect was only observed at the Zhangye and Teheran stations, showing $\delta^{18}O$-precipitation gradients of 0.20‰/mm and −0.08‰/mm, respectively. Latitude and altitude were always significantly correlated with annual $\delta^{18}O$, when considering geographical controls on $\delta^{18}O$, with $\delta^{18}O$/LAT and $\delta^{18}O$/ALT gradients of −0.42‰/° and −0.001‰/m, respectively. But both latitude and longitude were significantly correlated with $\delta^{18}O$ in winter. The relationship between $\delta^{18}O$ and large-scale atmospheric circulation suggested that the moisture in Central Asia is mainly transported by westerly circulation and is indirectly affected by the Indian monsoon. Meanwhile, the East Asian monsoon may affect the precipitation $\delta^{18}O$ content in westerly and monsoon transition regions. These results improve our understanding of the precipitation $\delta^{18}O$ and moisture transport in Central Asia, as well as the paleoclimatology and hydrology processes in Central Asia.

Corresponding authors
Junqiang Yao, yaojq@idm.cn
Xinchun Liu, liuxch@idm.cn

## INTRODUCTION

Stable water isotopes, including $\delta^{18}O$, and $\delta^2H$, are critical indicators of global and regional water cycles and paleoclimatic investigations improve our understanding of hydrological and atmospheric processes (*Dansgaard, 1964*; *Craig & Gordon, 1965*; *Song et al., 2007*; *Sophocleous, 2002*; *Zhang et al., 2004*; *Yao et al., 2013*). Precipitation is a critical variable of the global hydrological cycles. The isotope composition of precipitation from different sources varies and can be used as natural tracers to determine the sources and moisture

transport of water vapor as well as water cycle processes in different climatic regions (*Yamanaka et al., 2007*; *Li et al., 2012*; *Dansgaard, 1953*; *Chen, 2014*; *Zhang & Wang, 2016*).

In 1961, the International Atomic Energy Agency (IAEA), in conjunction with the World Meteorological Organization (WMO), initiated the Global Network of Isotopes in Precipitation (GNIP) on a global scale to survey the stable hydrogen and oxygen isotope and tritium composition in precipitation. The number of observation stations has since gradually increased. In terms of the spatial distribution, the GNIP observation stations are mainly concentrated in low-latitude wet regions, with few stations inland and in arid regions. For example, there are only ten GNIP stations that record $\delta^{18}$O in Central Asia and surrounding regions, including three stations in the arid region of Northwest China (i.e., the Urumqi, Hetian, and Zhangye stations) (*Gu, 2012*). However, most recording stopped in the mid-1990s (*Liu et al., 2014*) and information on the stable isotope composition of precipitation in Central Asia and the surrounding regions is limited.

Since the 1980s, China has become expert in the observation and investigation of stable isotope compositions in precipitation, mostly focusing on the Tibetan Plateau (*Gao et al., 2011*; *Tian et al., 1997*; *Tian et al., 2001*; *Yao et al., 2006*). Similar to the GNIP database, the Tibetan Network for Isotopes in Precipitation (TNIP) was established to observe and monitor the Tibetan Plateau and the surrounding regions; the China Network of Isotopes in Precipitation and Rivers (CHIRP) was developed for nationwide observation (*Yao, 2009*; *Liu et al., 2014*). In 2004, the Chinese Network of Isotopes in Precipitation (CHNIP) was started by the Chinese Ecosystem Research Network (CERN) (*Song et al., 2007*). However, there were only two CHNIP stations in Northwest China (the Fukang and Cele stations). An observation network was established in 2012 around the Tianshan Mountains (TSNIP) to investigate the types of precipitation isotopes in the high mountains over Central Asia (*Wang et al., 2016*).

Few studies have investigated precipitation isotopes around the Tibetan Plateau and Northwest China. Using observations and simulations, *Yao et al. (2013)* systematically examined the spatial-temporal distribution of precipitation $\delta^{18}$O over the Tibetan Plateau and determined its relationship with the mechanisms of the westerlies, Indian monsoons, and transitions in between.

Central Asia is one of the largest arid regions in the world and includes five Central Asian countries and the arid region of Northwest China. The topography is complex, with desert and oases coexisting. Precipitation is the main source of water for the mountainous regions but is unevenly distributed (*Chen, 2012*; *Yao et al., 2014*). The moisture source, transport path, and atmospheric processes are complex and are particularly sensitive to climate change in the unique mountain-basin structure of Central Asia (*Chen, 2014*). Few studies have reported the changes and mechanisms of moisture transport processes in Central Asia (*Huang et al., 2015*, *2018*; *Shi & Sun, 2008*; *He et al., 2016*; *Chen et al., 2010*; *Yao et al., 2020*). Previous studies have mainly focused on the effects of spatial-temporal characteristics, compositions, and climatic controls on precipitation isotopes on a regional scale, such as in the Tianshan Mountains and at Urumqi station (*Yao et al., 1999*; *Kong, Pang & Froehlich, 2013*; *Kong & Pang, 2016*; *Liu et al., 2015*; *Wang et al., 2016*, *2017*;

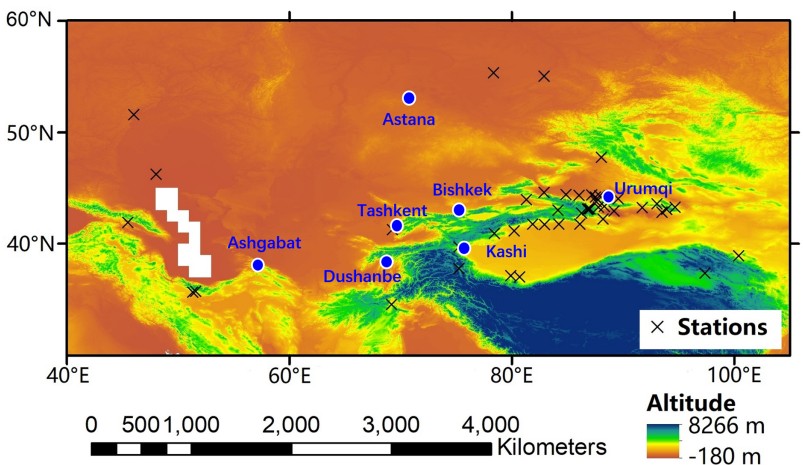

**Figure 1** Locations of the sampling stations over Central Asia.

*Zhang & Wang, 2018*). However, only a few studies have been conducted on the composition of stable isotopes in the precipitation over the Central Asian region.

Central Asia has a unique water cycle and complex moisture transport processes. We compiled a database of information from 47 stations to evaluate precipitation $\delta^{18}$O over Central Asia to reveal its spatial-temporal characteristics. We determined the relationships between precipitation $\delta^{18}$O and local factors, such as geography (latitude and altitude) and climatic parameters (temperature and precipitation). We also investigated the relationship between $\delta^{18}$O and large-scale atmospheric circulation.

## MATERIALS & METHODS

We studied the precipitation isotopes data ($\delta^{18}$O and $\delta^2$H) from Central Asia to comprehensively evaluate the processes and characteristics of the composition of the stable isotopes in the precipitation of that region.

From the observed monthly precipitation, $\delta^{18}$O and $\delta^2$H was obtained from 47 observation stations over Central Asia, including 12 stations of the GNIP, 22 stations of the TSNIP (*Wang et al., 2016*), two TNIP stations (*Yao et al., 2013*), two CHNIP stations (*Liu et al., 2014*), and nine stations from reference studies (Fig. 1 and Table 1).

The Kabul station had the longest data record, with monthly data starting in the 1960s and continuous data for 1962–1964, 1967–1975, and 1982–1989. The Urumqi station kept monthly data starting in 1986 and had continuous data for 1961–1976, 1979–1981, 1986–1987, and 2000–2004. Among the stations from which data was collected, five GNIP stations (Teheran, Kabul, Astrakhan, Urumqi, and Zhangye stations) took samples for more than 10 years and data were available for interannual variability studies (Fig. 1). Furthermore, eight GNIP stations (Teheran, Kabul, Saratov, Barabinsk, Astrakhan, Urumqi, Hetian, and Zhangye stations) were sampled for more than 30 months and data were available to evaluate the relationship of precipitation $\delta^{18}$O with temperature and precipitation over Central Asia (Fig. 1). Spatial patterns of observed precipitation $\delta^{18}$O over Central Asia were available from all stations. All precipitation $\delta^{18}$O data were

**Table 1  Summary data of the precipitation sampling stations over Central Asia.**

| No | Station | Latitude (°N) | Longitude (°E) | Altitude (m) | Precipitation (mm) | Temperature (°C) | Database |
|----|---------|---------------|----------------|--------------|--------------------|------------------|----------|
| 1 | Urumqi | 43.78 | 87.62 | 918 | 304 | 7.4 | GNIP |
| 2 | Hetian | 37.13 | 79.93 | 1375 | 209 | 9.1 | GNIP |
| 3 | Zhangye | 38.93 | 100.43 | 1483 | 154 | 7.8 | GNIP |
| 4 | Kabul | 34.57 | 69.21 | 1860 | 330 | 11.6 | GNIP |
| 5 | Barabinsk | 55.33 | 78.37 | 120 | 389 | 1 | GNIP |
| 6 | Astrakhan | 46.25 | 48.03 | −18 | 228 | 10.5 | GNIP |
| 7 | Saratov | 51.56 | 46.03 | 166 | 490 | 6.6 | GNIP |
| 8 | Teheran | 35.68 | 51.32 | 1,200 | 227 | 16.7 | GNIP |
| 9 | Tashkent | 41.27 | 69.27 | 428 | 478 | 13.9 | GNIP |
| 10 | Teheran East | 35.74 | 51.58 | 1,350 | 297 | 15.4 | GNIP |
| 11 | Novosibirsk | 55.03 | 82.9 | 162 | 479 | 2.5 | GNIP |
| 12 | Telavi | 41.93 | 45.48 | 590 | 679 | 13.1 | GNIP |
| 13 | Taxkorgen | 37.77 | 75.27 | 3,100 | 115 | 1.6 | TNIP |
| 14 | Delingha | 37.37 | 97.37 | 2,981 | 186 | 2.2 | TNIP |
| 15 | Fukang | 44.29 | 87.93 | 460 | 167 | 7.5 | CHNIP |
| 16 | Cele | 37.02 | 80.73 | 1,306 | 51 | 12.9 | CHNIP |
| 17 | Yining | 43.95 | 81.33 | 662.5 | 298.9 | 9.5 | TSNIP |
| 18 | Jinghe | 44.62 | 82.9 | 320.1 | 112.1 | 8.2 | TSNIP |
| 19 | Kuytun | 44.4 | 84.87 | 562 | 183.5 | 8.5 | TSNIP |
| 20 | Shihezi | 44.32 | 86.05 | 442.9 | 226.9 | 7.8 | TSNIP |
| 21 | Caijiahu | 44.2 | 87.53 | 440.5 | 153.8 | 6.5 | TSNIP |
| 22 | Qitai | 44.02 | 89.57 | 793.5 | 200.9 | 5.4 | TSNIP |
| 23 | Wuqia | 39.72 | 75.25 | 2,175.7 | 188.7 | 7.7 | TSNIP |
| 24 | Akqi | 40.93 | 78.45 | 1,984.9 | 237.7 | 6.8 | TSNIP |
| 25 | Bayinbuluke | 43.03 | 84.15 | 2,458 | 280.5 | −4.2 | TSNIP |
| 26 | Baluntai | 42.73 | 86.3 | 1,739 | 220.4 | 7 | TSNIP |
| 27 | Balikun | 43.57 | 93.05 | 1,677.2 | 230.5 | 2.7 | TSNIP |
| 28 | Yiwu | 43.27 | 94.7 | 1,728.6 | 104.4 | 4.2 | TSNIP |
| 29 | Aksu | 41.17 | 80.23 | 1,103.8 | 80.4 | 10.8 | TSNIP |
| 30 | Baicheng | 41.78 | 81.9 | 1,229.2 | 136.6 | 8.2 | TSNIP |
| 31 | Kuqa | 41.72 | 82.97 | 1,081.9 | 76.7 | 11.3 | TSNIP |
| 32 | Luntai | 41.78 | 84.25 | 976.1 | 78.6 | 11.6 | TSNIP |
| 33 | Korla | 41.75 | 86.13 | 931.5 | 59.2 | 12 | TSNIP |
| 34 | Kumux | 42.23 | 88.22 | 922.4 | 59.9 | 9.8 | TSNIP |
| 35 | Dabancheng | 43.35 | 88.32 | 1,103.5 | 76.7 | 6.9 | TSNIP |
| 36 | Turpan | 42.93 | 89.2 | 34.5 | 15.4 | 15.1 | TSNIP |
| 37 | Shisanjianfang | 43.22 | 91.73 | 721.4 | 22.6 | 12.5 | TSNIP |
| 38 | Hami | 42.82 | 93.52 | 737.2 | 43.7 | 10.3 | TSNIP |
| 39 | UG1 | 43.1 | 86.84 | 3,693 | 460 | −5.6 | TGS |
| 40 | Zongkong | 43.11 | 86.89 | 3,404 | 400 | −4.6 | TGS |
| 41 | Daxigou | 43.11 | 86.86 | 3,539 | 458 | −5 | TGS |

| No | Station | Latitude (°N) | Longitude (°E) | Altitude (m) | Precipitation (mm) | Temperature (°C) | Database |
|---|---|---|---|---|---|---|---|
| Table 1 (continued) | | | | | | | |
| 42 | Altay | 47.73 | 88.08 | 735 | 193 | 4.5 | *Tian et al., 2001* |
| 43 | Gaoshan | 43.1 | 86.83 | 3,545 | 390 | −4.3 | *Kong, Pang & Froehlich, 2013* |
| 44 | Houxia | 43.28 | 87.8 | 2,100 | 424 | 1.5 | *Kong, Pang & Froehlich, 2013* |
| 45 | Yuejinqiao | 43.12 | 87.05 | 2,526 | 330 | −2.5 | *Yao et al., 1999* |
| 46 | Yingxiongqiao | 44.37 | 87.2 | 1,920 | 210 | 2.3 | *Sun et al., 2015* |
| 47 | Yushugou | 43.08 | 93.95 | 1,670 | 94.3 | 18.6 | *Wang et al., 2015* |

Note:
GNIP, Global Network of Isotopes in Precipitation; TNIP, Tibetan Network for Isotopes in Precipitation; CHNIP, Chinese Network of Isotopes in Precipitation; TSNIP, Tianshan Network of Isotopes in Precipitation; and TGS, Tianshan Glaciological Station, Chinese Academy of Sciences.

employed with respect to the Vienna Standard Mean Ocean Water (VSMOW) and were shown in precipitation amount-weighted values. In addition, meteorological variables, including precipitation amount and air temperature, were recorded at each observation station.

The precipitation $\delta^{18}O$ was determined by atmospheric characteristics and their dynamics. We selected three atmospheric characteristic indices to discuss their effects on the precipitation $\delta^{18}O$ over Central Asia, including the westerly circulation index (WCI), East Asian summer monsoon index (EASMI), and Indian monsoon index (IMI). Monthly EASMI and IMI were obtained from Dr. Jianping Li's webpage (http://ljp.gcess.cn/dct/page/65610) (*Li & Zeng, 2002, 2003*).

Temperature averages are calculated for monthly/annual analyses, and the corresponding precipitation amounts are calculated as monthly/annual data for each observation station. A stepwise linear regression analysis technique was employed to fit the precipitation $\delta^{18}O$ with geographical parameters. The following geographical parameters were considered: altitude (ALT, m), latitude (LAT, °N), and longitude (LON, °E). The Pearson correlation coefficient was used to investigate the relationship between precipitation $\delta^{18}O$ and meteorological variables or atmospheric circulation indices.

# RESULTS AND DISCUSSION

## Local meteoric water lines (MWL)

The local meteoric water lines (MWLs) of precipitation, $\delta^2H$, and $\delta^{18}O$, provide important information on the water cycle, water vapor sources, and water transport (*Jonesl, Leng & Arrowsmith, 2007*; *Lutz, Thomas & Panorska, 2011*). A Central Asia MWL (CAMWL) was established as $\delta^2H = 7.30\delta^{18}O + 3.12$ ($R^2 = 0.95$, $n = 727$, $p < 0.01$, R is the Pearson correlation coefficient) (Fig. 2) based on 727 precipitation groups. The slope of the CAMWL was slightly lower than that of the global MWL (GMWL), which was eight (*Craig, 1961*) and the Chinese MWL of 7.48 (*Liu et al., 2014*). Differences in the local MWLs slope often occur because of deviations in humidity at the source of moisture or evaporation. Central Asia is located in the Eurasian hinterland, and has a significant variation in the annual cycle and alternating dry or wet seasons. A low LMWL slope is associated with non-equilibrium conditions that affect falling raindrops during dry conditions (*Liu et al., 2014*), leading to the potential for significant sub-cloud evaporation.

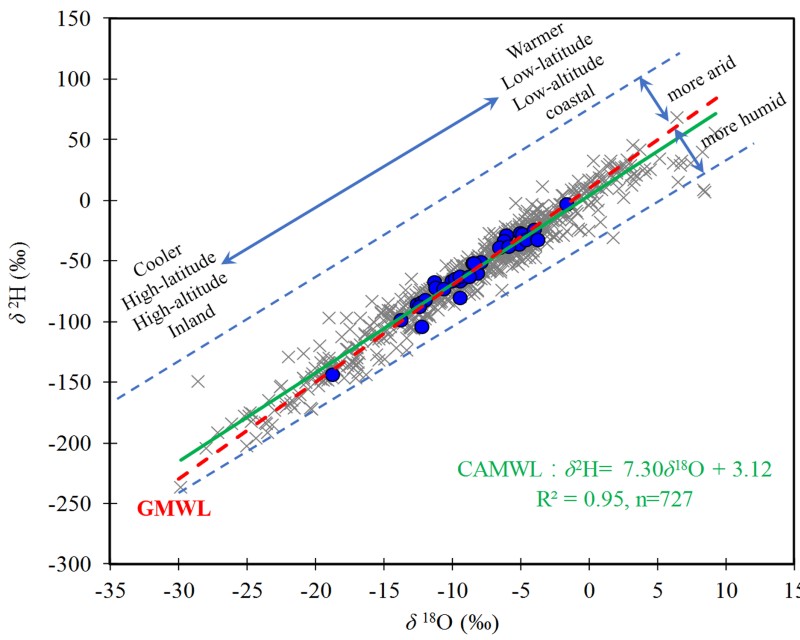

**Figure 2 Linear $\delta^2H$–$\delta^{18}O$ relationships (CAMWL) based on all precipitation measurements over Central Asia.**

The precipitation $\delta^{18}O$ over Central Asia ranged from +2‰ to −25.4‰ with a mean of −8.7‰, and $\delta^2H$ ranged from −4.2 ‰ to −191.4‰ with a mean of −67.1 ‰.

## Seasonal variations in precipitation $\delta^{18}O$

The arid Central Asian region is affected by monsoons and westerlies, resulting in annual differences in climatic variables and precipitation $\delta^{18}O$. The variation in the annual air temperature represents a continental climate. The maximum and minimum temperatures occur in July and January, respectively. In this study, the temperature variation was unimodal, and the precipitation amount represented three distribution types (Fig. 3). The three precipitation patterns observed in different regions are shown in Fig. 3. (1) Maximum precipitation in occurred in the summer, with summer precipitation accounting for 41.4% of the total precipitation. These stations (including Hetian, Zhangye, Barabinsk, and Urumqi stations) are mainly distributed in northern Central Asia and are primarily influenced by the intensity and location of the westerly circulation. (2) Maximum precipitation values in winter and spring, with winter and spring precipitation accounting for 39.6% and 37.8% of the total precipitation, respectively, while the summer precipitation accounted for only 6.4% of the total precipitation. These stations (including the Kabul and Teheran stations) are mainly located in southern Central Asia and are primarily influenced by the Indian monsoon. (3) A well-distributed seasonal precipitation pattern was observed at Astrakhan and Saratov stations. These are present in northwestern Central Asia and are primarily influenced by airflow from the Arctic. This result also highlighted the complexity of the spatial–temporal variations in precipitation in Central Asia.

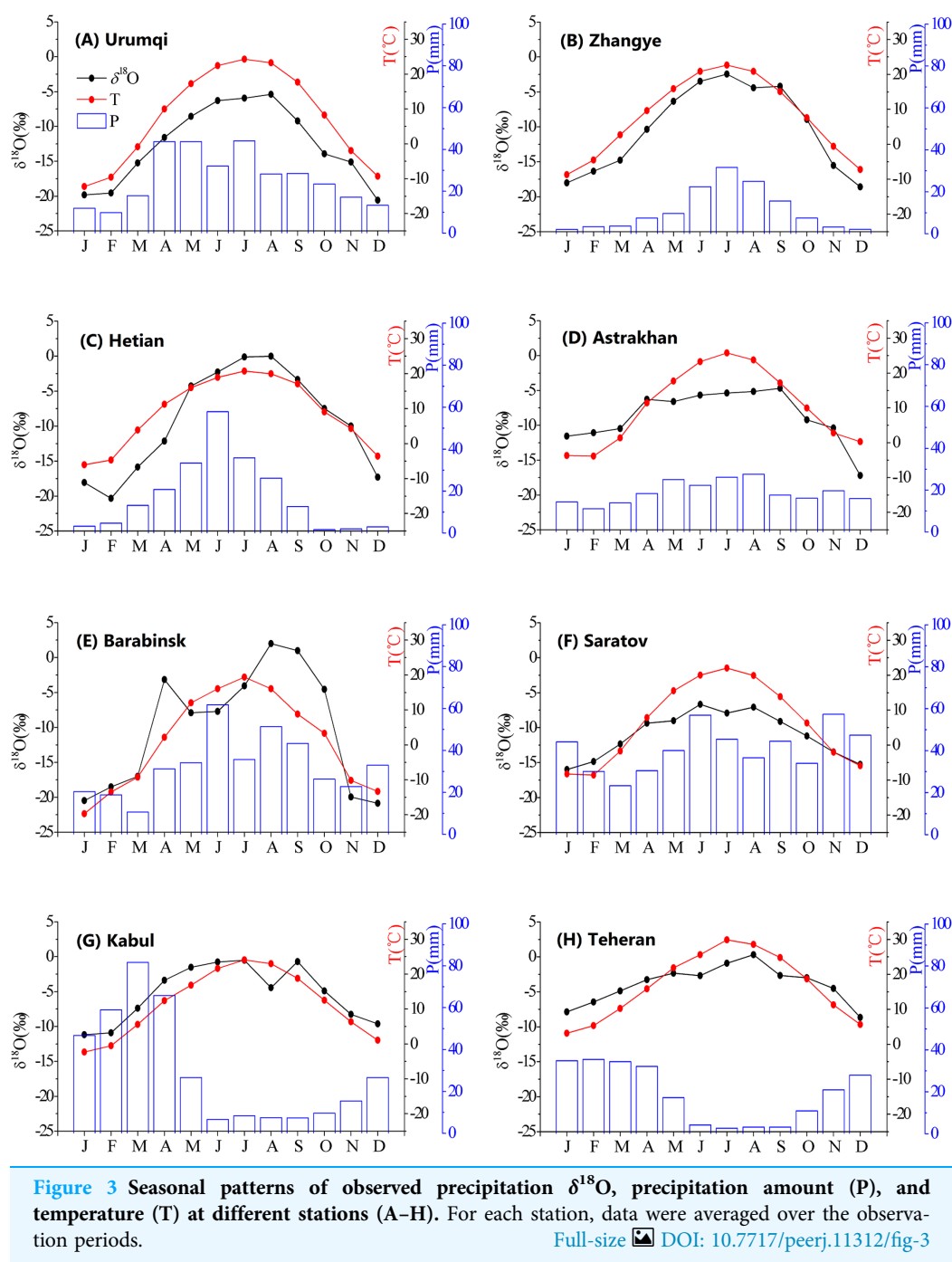

**Figure 3 Seasonal patterns of observed precipitation $\delta^{18}O$, precipitation amount (P), and temperature (T) at different stations (A–H).** For each station, data were averaged over the observation periods.

The annual temperature and precipitation influenced the annual precipitation $\delta^{18}O$ cycle. The maximum $\delta^{18}O$ occurred from June to August (JJA). Maximum precipitation $\delta^{18}O$ at the Urumqi, Hetian, Barabinsk, and Teheran stations were −5.4‰, 0‰, 2‰, and −0.3‰ in August, respectively. The maximum values at the Zhangye and Kabul stations occurred in July. The maximum values occurred in June and September, respectively, at the Saratov and Astrakhan stations. All minimum values occurred from December to February (DJF). Therefore, the $\delta^{18}O$ content in precipitation was higher in

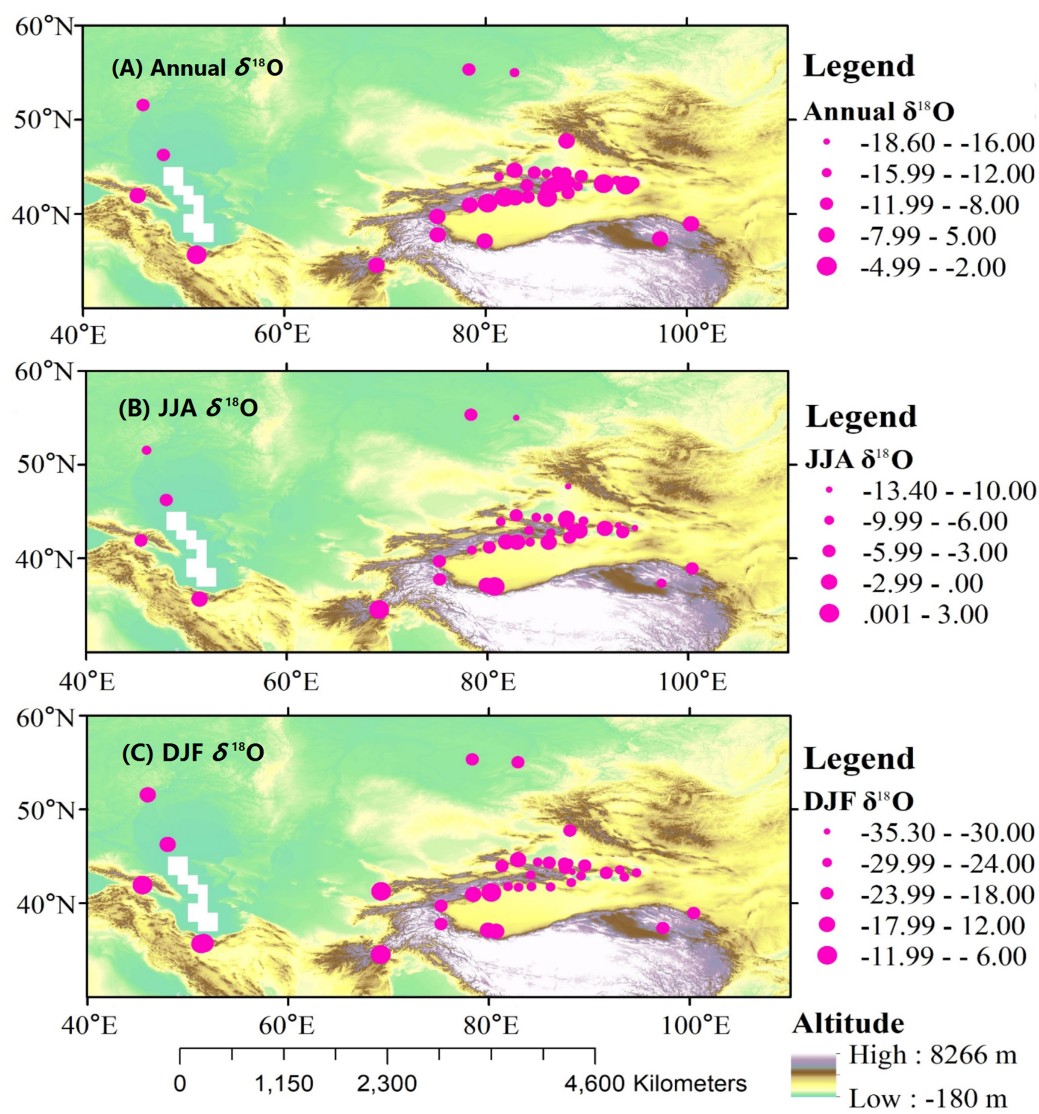

**Figure 4 Spatial patterns of annual (A), JJA (B), and DJF (C) observed precipitation $\delta^{18}O$ at meteorological stations over Central Asia.**

summer than in winter, and the seasonal pattern of the $\delta^{18}O$ in precipitation reflected the climate regime during the annual cycle. In winter, air masses were relatively cold and dry and the amount of precipitation was very low, while in summer, the opposite was true.

## Spatial characteristics of precipitation $\delta^{18}O$

Figure 4 shows the spatial distribution of precipitation $\delta^{18}O$ throughout the year, and for summer and winter seasons at each station in Central Asia. The maximum value of the annual mean $\delta^{18}O$ (−1.47‰) was at Cele station, located in southern Xinjiang, and the minimum value (−18.6‰) was observed at Hami station. For the summer months, the maximum $\delta^{18}O$ value (2.8 ‰) was observed at Cele station, and the minimum value (−13.4‰) was observed at Altay station, located in northeastern Central Asia. The maximum $\delta^{18}O$ value (−7.7 ‰) during the winter months was observed at the

Teheran station, located in southern Central Asia, and the minimum value (−35.3‰) was observed at the Dabancheng station.

There were remarkable spatial differences in precipitation $\delta^{18}$O over Central Asia. The spatial distribution of precipitation $\delta^{18}$O was mainly affected by the thermodynamics of water vapor condensation during the Rayleigh fractionation process and included meteorological elements, water vapor transport, and geographic factors (*Dansgaard, 1964*; *Yurtsever & Gat, 1981*; *Rozanski, Araguas-Araguas & Gonfiantini, 1993*).

Several studies have verified that altitude and latitude are the main geographic factors that affect changes in temperature and water vapor condensation, and can be referred to as geographic factor effects of precipitation $\delta^{18}$O (*Liu, Tian & Yao, 2009*; *Yao et al., 2013*). The correlation of precipitation $\delta^{18}$O with latitude, altitude, and longitude was investigated using partial correlation analysis, and the R were 0.53 ($p < 0.05$), 0.32 ($p < 0.05$), and 0.17 ($p > 0.05$), respectively, indicating that latitude is the most significant factor. In summer, latitude was significantly correlated with precipitation $\delta^{18}$O (R = 0.54, $p < 0.05$), while both latitude and longitude were significantly correlated with precipitation $\delta^{18}$O in winter (R = 0.31, $p < 0.05$; R = 0.67, $p < 0.01$, respectively). For the annual $\delta^{18}$O, the $\delta^{18}$O/LAT gradient was −0.42‰/°, implying that the precipitation $\delta^{18}$O was reduced by approximately 0.42‰ for every one-degree variation in latitude, which was larger than the gradient in China (−0.22‰/°). Similarly, the $\delta^{18}$O/ALT gradient was −0.001‰/m, which was lower than the global value of −0.0022‰/m and the value of −0.0016‰/m in China (*Liu, Tian & Yao, 2009*; *Bowen & Wilkinson, 2002*). In Central Asia, altitudes range from −18 to 4,200 m, with a complex topography and various climatic characteristics.

Several researchers have constructed models to explain the relationship between precipitation $\delta^{18}$O and geographic factors (*Liu, Tian & Yao, 2009*; *Yao et al., 2013*), including the Bowen and Wilkinson (BW) model (*Bowen & Wilkinson, 2002*). To confirm the effects of altitude and latitude on precipitation $\delta^{18}$O over Central Asia, we evaluated the relationship between precipitation $\delta^{18}$O and geographical factors using stepwise regression analysis.

In Central Asia, geographical controls on precipitation $\delta^{18}$O were best expressed by a stepwise linear regression, including altitude (*ALT*, m), latitude (*LAT*, °N) and longitude (*LON*, °E):

$$\delta^{18}O = (-0.179) \times LAT + (-0.022) \times LON + 0.00085 \times ALT \ (R^2 = 0.51, p < 0.05) \quad (1)$$

The best model accounting for JJA $\delta^{18}$O included both altitude (*ALT*, m) and latitude (*LAT*, °N):

$$\delta^{18}O = (-0.632) \times LAT + (-0.002) \times ALT + 23.851 \ (R^2 = 0.64, p < 0.05) \quad (2)$$

The DJF $\delta^{18}$O model was expressed by linear regression as:

$$\delta^{18}O = (-0.482) \times LAT + (-0.280) \times LON + (-0.001) \times ALT \\ + 23.891 (R^2 = 0.73, p < 0.01) \quad (3)$$

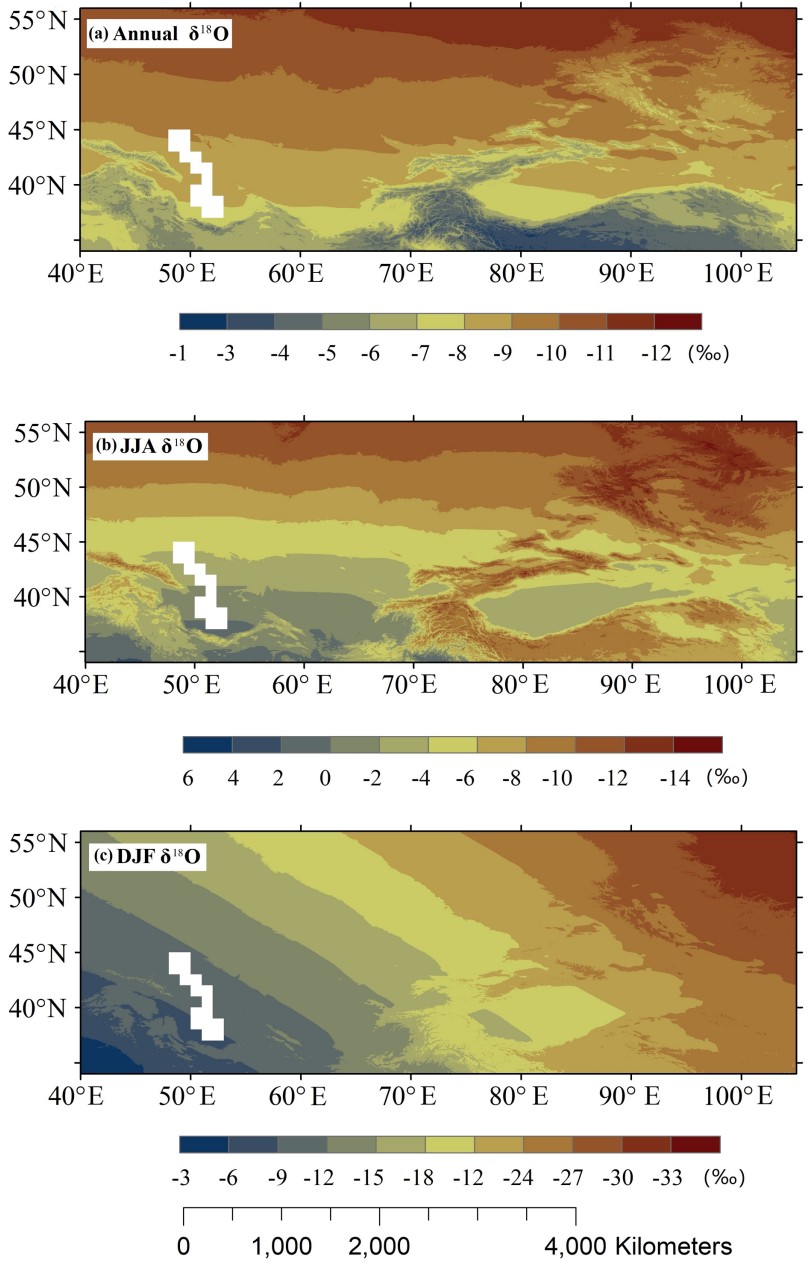

**Figure 5 Spatial distribution of annual (A), JJA (B), and DJF (C) mean estimated $\delta^{18}O$ in precipitation based on the model using spatial variables.**

Our model comprehensively explained the effect of altitude, latitude, and longitude on precipitation $\delta^{18}O$ over Central Asia, and revealed that the variation in precipitation $\delta^{18}O$ depended on geographical factors at each station. These results showed that our models were suitable for use in Central Asia.

We used the aforementioned models to obtain the spatial distribution of precipitation $\delta^{18}O$ over Central Asia (Fig. 5). The annual precipitation $\delta^{18}O$ gradually decreased as the latitude increased (Fig. 5A). In the summer, higher $\delta^{18}O$ values were observed in

southwest Central Asia and lower $\delta^{18}$O values were observed in northern Central Asia and the Pan-Third Polar region (including the Tibetan Plateau, Iranian Plateau, and Tianshan Mountains) (Fig. 5B). This pattern shows the effects of latitude and altitude on precipitation $\delta^{18}$O. However, precipitation $\delta^{18}$O in Central Asia was higher than that in the Tibetan Plateau (Fig. 5B). This was mainly due to the transport of moist air originating from the Arabian Sea and the Bay of Bengal into Central Asia, which resulted in more summer precipitation (*Zhao et al., 2014*). A strong anti-cyclonic pattern was noted in the Arabian Sea, Indian subcontinent, and the Bay of Bengal, and was associated with a strong southerly flow controlling the Indian subcontinent and extending north up to the valley between the Iranian Plateau and Tibetan Plateau regions. Furthermore, the valley between the Iranian Plateau and the Tibetan Plateau is below 1,500 m, and the anomalous southerlies can transport moisture into Central Asia (*Zhao et al., 2014*; *Zhao & Zhang, 2015*). The cyclonic pattern observed in Central Asia was associated with this pattern, and an anomalous southwesterly wind may have transported moisture into Central Asia. *Tian et al. (2001)* indicated that there was a different moisture source between the northern and southern Tibetan Plateau, and that the northern limit of the summer monsoon was north of the Yarlung Zangbo River located in the middle of the Tibetan Plateau.

In the winter, precipitation $\delta^{18}$O gradually decreased with increasing longitude (Fig. 5C). The $\delta^{18}$O values in Central Asia was lower than that in the eastern Asia and Mongolian Plateau. The moisture transported from the Eurasian continent and the Mediterranean Sea into northern Central Asia generated more winter precipitation and westerly wind was needed to transport moist air into inland regions.

## Correlation between precipitation $\delta^{18}$O and meteorological variables

We investigated correlations between monthly precipitation $\delta^{18}$O, monthly air temperature (Fig. 6), and precipitation at eight stations in Central Asia (Fig. 7). As shown in the figure, monthly precipitation $\delta^{18}$O at each station had a significant positive correlation with temperature (R = 0.51–0.96, $p < 0.01$). Moreover, the latitude effect with $\delta^{18}$O gradually increased as the temperature increased. The effect of temperature was the strongest at the Barabinsk station, followed by those at the Urumqi, Hotan, Zhangye, Saratov, and Kabul stations, while it was the weakest at the Astrakhan and Teheran stations. The gradients between $\delta^{18}$O and air temperature ranged from 0.28‰/°C at the Saratov and Teheran stations to 0.68‰/°C at the Hetian station. These results were similar to global mid- and high-latitudes with gradients of 0.55‰/°C (*Rozanski, Araguas-Araguas & Gonfiantini, 1993*) and elsewhere in China with gradients of 0.36‰/°C (*Gu, 2011*).

Significant positive correlations between precipitation $\delta^{18}$O and precipitation were observed at four stations (Zhangye, Barabinsk, Hetian, and Urumqi), and only one station had a significant negative correlation (Teheran station). Correlations for other stations were not significant. These results reveal an indeterminant dependence of precipitation $\delta^{18}$O on precipitation, which is in accordance with the insignificant effect of precipitation in inland regions (*Gu, 2012*). The $\delta^{18}$O–precipitation gradients ranged from −0.08‰/mm (Teheran station) to 0.20 ‰/mm (Zhangye station). In addition, the Zhangye and

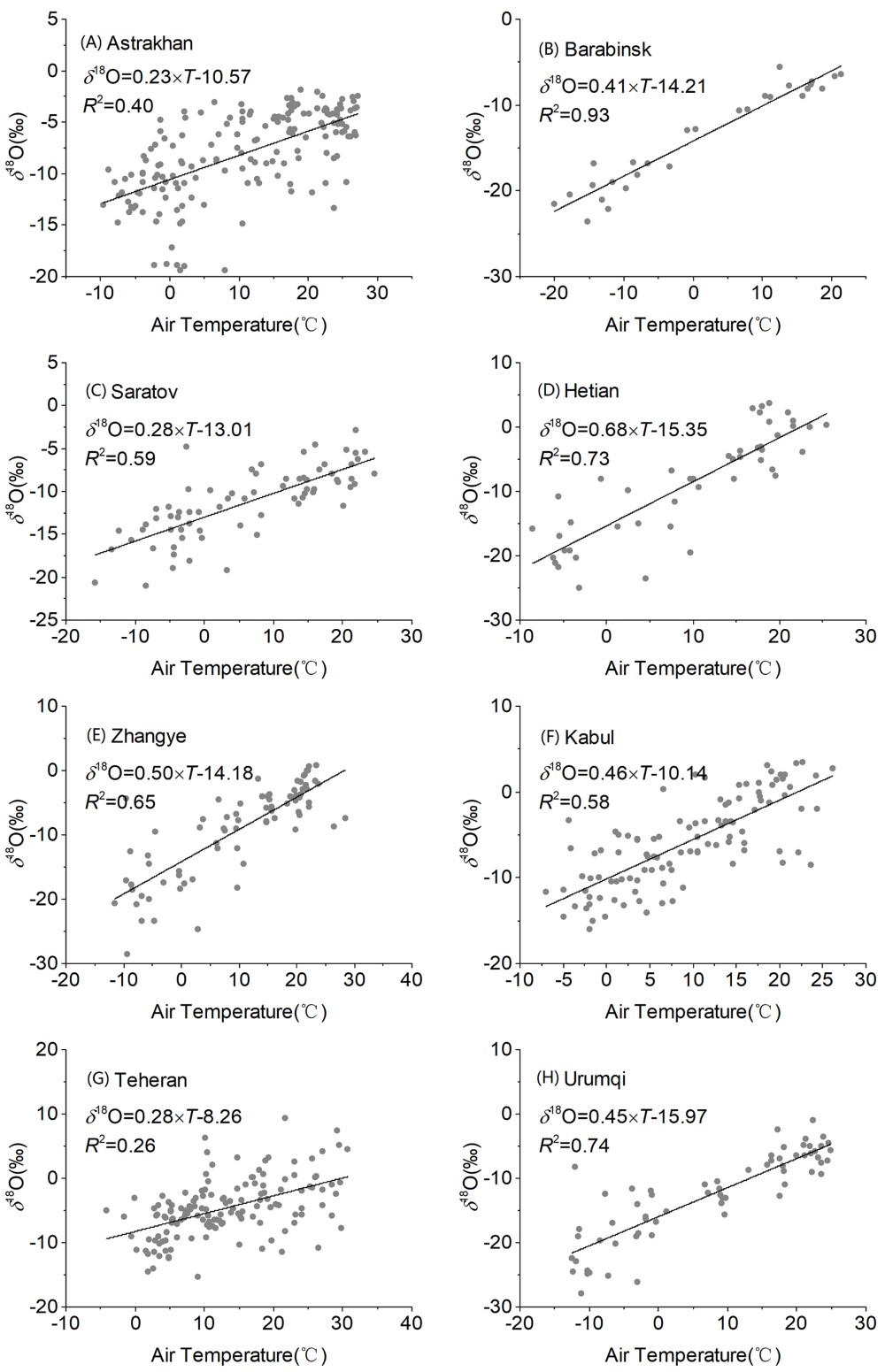

**Figure 6 Regression of observed precipitation δ¹⁸O with temperature over Central Asia at different stations (A–H).**

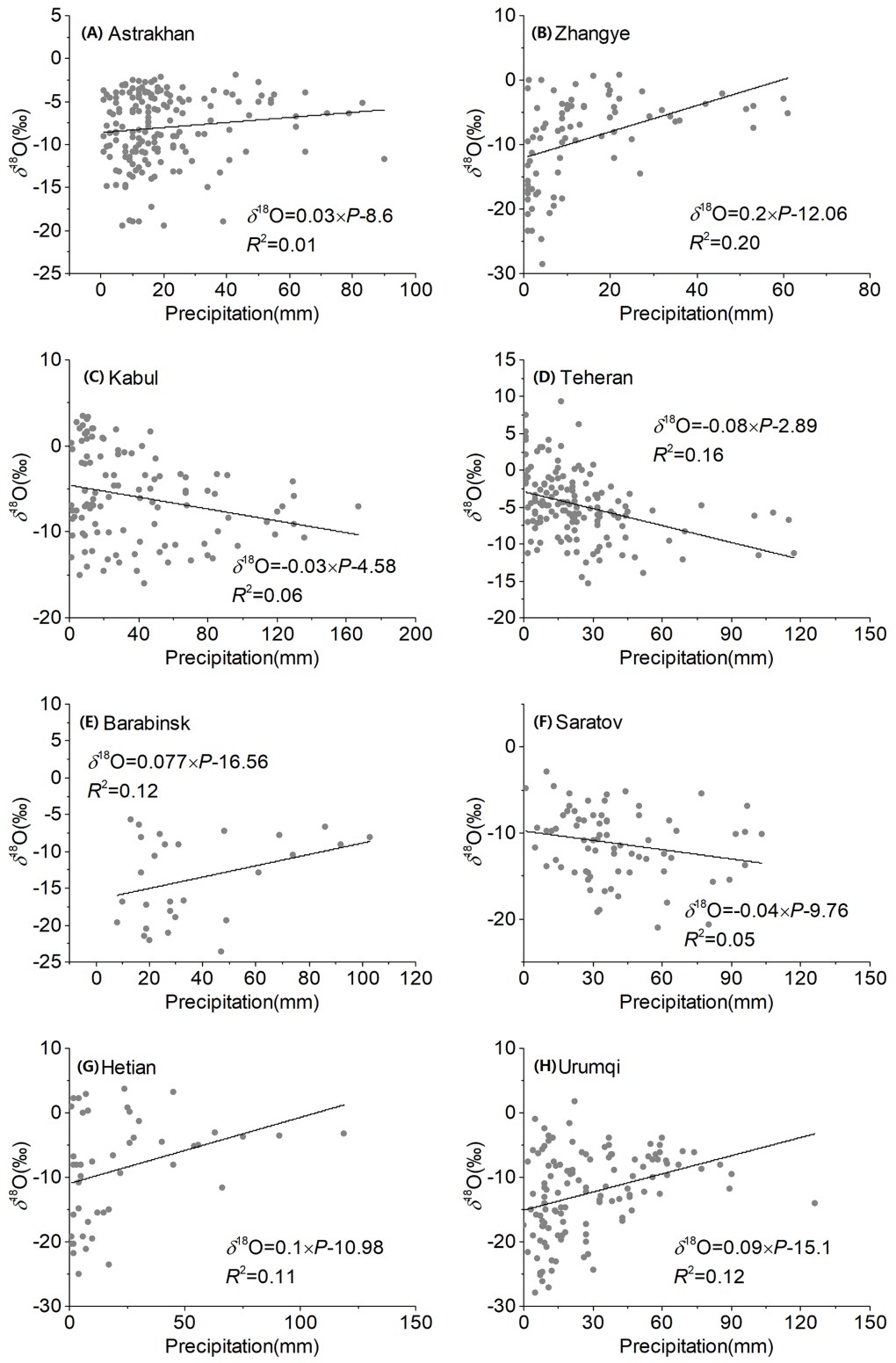

**Figure 7 Regression of observed precipitation δ¹⁸O with amount of precipitation over Central Asia at different stations (A–H).**
Teheran stations had a significant precipitation effect as a result of their location on the edge of the East Asian monsoon and Indian monsoon regions, respectively.

The Urumqi station has the longest and most systematic GNIP observations in Central Asia, with an observation period spanning from 1986 to 2003. Based on a subset of selected predictors, we established a model using stepwise regression analysis and the corresponding monthly temperature and precipitation data. The $\delta^{18}$O model in Urumqi station was expressed by linear regression as:

$$\delta^{18}O = (-0.009) \times P + 0.417 \times T - 15.53 \ (R^2 = 0.69, p < 0.01) \qquad (4)$$

We reconstructed and estimated the monthly precipitation $\delta^{18}$O time series from 1985 to 2013 at Urumqi station (Fig. 8A). The reconstructed precipitation $\delta^{18}$O was high-positively correlated with the observed values at Urumqi stations (R = 0.83, $p < 0.01$), and the results clearly depicted the seasonal cycle of precipitation $\delta^{18}$O, except for some values during extreme cold or hot months. Furthermore, the reconstructed precipitation $\delta^{18}$O values for most spring and autumn seasons are similar to the observations. This indicates that basic fractionation mechanisms can be reflected using the reconstructed model. The reconstructed model indicates that the temperature effect of precipitation $\delta^{18}$O values was reflected in Urumqi Station. Several previous studies confirmed the temperature effect of precipitation isotopes existed in arid Central Asia (*Yao et al., 1999*, *2013*; *Wang et al., 2016*). Thus, the selected control factors and established regressions can be used to reconstruct the long-term variation in precipitation $\delta^{18}$O, and also act as a proxy of an historic environment. In High Asia, the climatic controls on precipitation isotopes was used as a base for ice core sampling (*Tian & Yao, 2016*). In Central Asia, the $\delta^{18}$O of ice cores were in good agreement with temperatures at nearby stations, which indicates its value in paleoclimate reconstruction using ice cores (*Tian et al., 2006*; *Zhang & Wang, 2018*).

## Relationship of precipitation $\delta^{18}$O with general atmospheric circulation

To reveal the relationship between precipitation $\delta^{18}$O and large-scale atmospheric circulation, we analyzed the correlation between precipitation $\delta^{18}$O at each station using long-term observation data and atmospheric circulation (e.g., IMI, SASMI, and WCI).

Our results showed that the precipitation $\delta^{18}$O at the Zhangye station was positively correlated with the EASMI (R = 0.62, $p < 0.05$), indicating that the East Asian summer monsoon has an important effect on precipitation $\delta^{18}$O in Zhangye. It is located on the eastern side of the arid northwest China and at the northeastern edge of the Tibetan Plateau as well as at the transition zone between the East Asian monsoon and westerlies. The East Asian monsoon can affect the precipitation $\delta^{18}$O in the westerly and monsoon transition regions.

The precipitation $\delta^{18}$O was negatively correlated with the IMI at both the Urumqi and Teheran stations (R = −0.57 and −0.41, respectively, $p < 0.05$), revealing an important effect of the Indian monsoon on precipitation $\delta^{18}$O. In general, the stronger the monsoon, the lower the $\delta^{18}$O value, and vice versa. The Teheran station is close to the Indian Ocean and is one of the main moisture paths for the Indian monsoon moving towards the north.

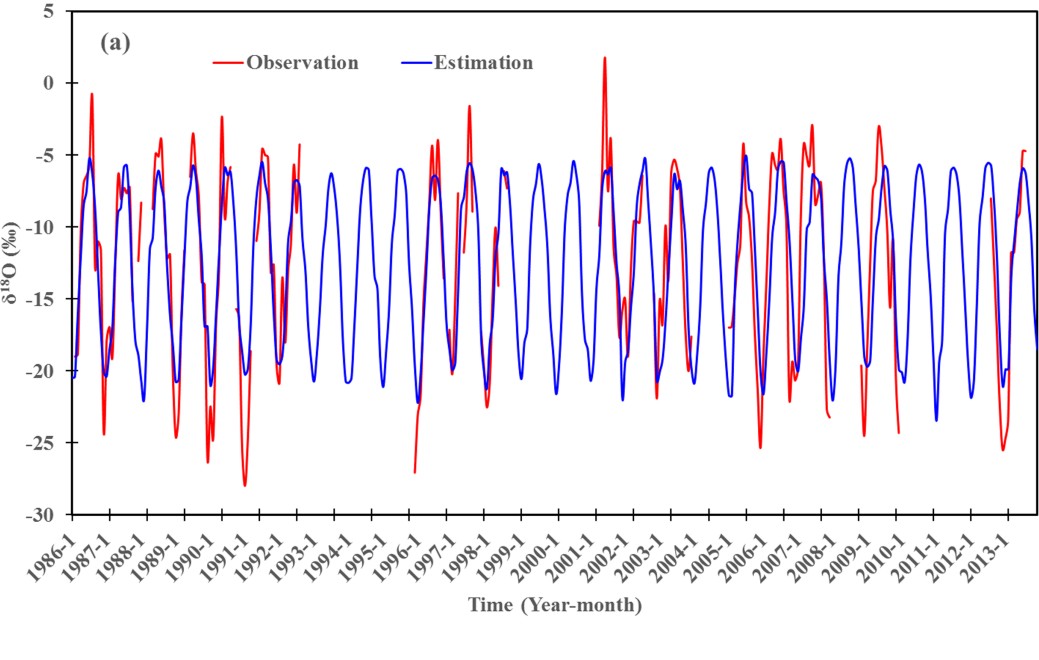

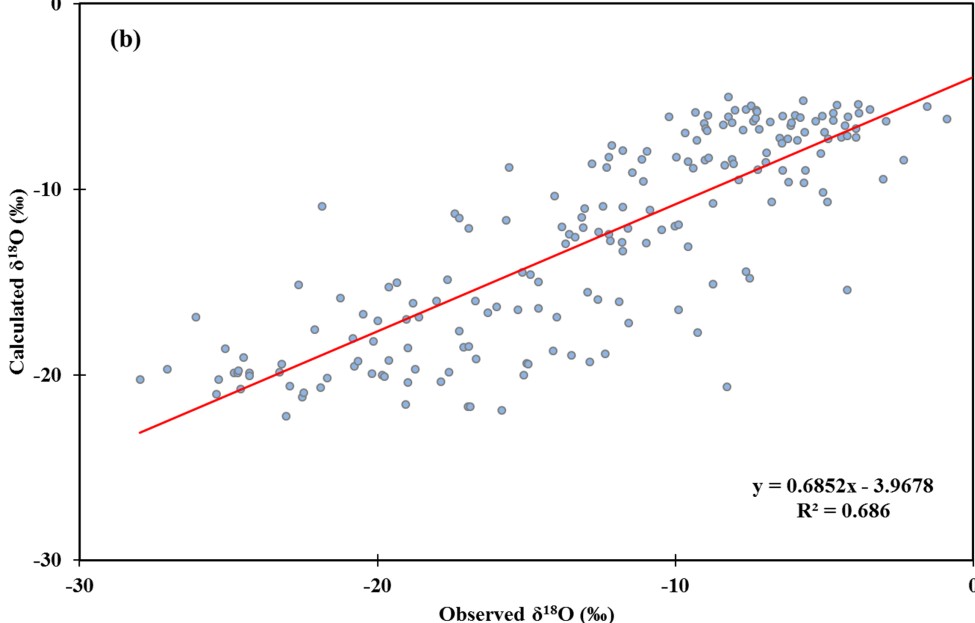

**Figure 8** (A) Reconstruction of monthly $\delta^{18}O$ time series during 1986–2013 based on the regression model established for Urumqi station. (B) Reconstructed values of monthly $\delta^{18}O$ plotted versus observed values.

Therefore, precipitation $\delta^{18}O$ at the Teheran station was mainly influenced by the Indian monsoon. However, Urumqi is located in the Asian hinterland and is indirectly influenced by the Indian monsoon. At Urumqi, precipitation $\delta^{18}O$ is affected by an anomalous moisture transport path through a multi-step process. A weakened Indian monsoon can cause an anomalous cyclone in the middle and upper troposphere in Central Asia, resulting in cooling, which is directly correlated with the increased precipitation

in Xinjiang (*Zhao et al., 2014*). Furthermore, the Indian monsoon affects moisture transport from the Bay of Bengal and the Arabian Sea to Xinjiang through a two-step process (*Zhao et al., 2014*). The extreme summer precipitation in northern Xinjiang was controlled by moisture sources originating from the Indian Ocean, which was closely related to a stronger meridional circulation (*Huang et al., 2017*). The anomalous circulation can also transport some moisture from the Indian Ocean along the eastern periphery of the Tibetan Plateau to North Xinjiang (*Huang et al., 2017*).

The moisture in Central Asia is mainly transported by the westerly circulation from the North Atlantic Ocean and the Eurasian continent. The westerly circulation is one of the most important factors affecting precipitation $\delta^{18}O$ over Central Asia. However, we found a weak positive correlation between precipitation $\delta^{18}O$ and the WCI index. This may be related to the moisture transported by the westerly circulation, which originates from the Atlantic and Arctic oceans and dissipates along the way. The water vapor transported by the westerly is exhausted by the time it reaches Central Asia because of substantial evapotranspiration and evaporation below the clouds in arid regions.

The precipitation d-excess parameter can provide supplementary information to precipitation $\delta^{18}O$ and $\delta^{2}H$ values, and it is largely controlled by the moisture source and water transport paths. Thus, it may be the most important indicator to represent the regional moisture source and atmospheric circulation (*Merlivat & Jouzel, 1979*). In addition, the precipitation d-excess is closely related to the re-evaporation of raindrops during condensation and precipitation, and is affected by temperature and relative humidity during evaporation (*Merlivat & Jouzel, 1979*; *Jouzel, Merlivat & Lorius, 1982*; *Jouzel, Froehlich & Schotterer, 1997*; *Wang, 2014*). In Central Asia, the precipitation d-excess values ranged from +21.66 ‰ to −8.5‰, with an average of 8.5 ‰. The temporal variations in precipitation d-excess shows more positive values in spring and more negative values in summer and the opposite pattern for $\delta^{18}O$. This variation is in agreement with variations in the conditions under which the moisture source evaporates. There is a similar seasonal pattern in precipitation d-excess and $\delta^{18}O$ of Central Asia and that of the westerly climate regime zone (*Yao, 2009*). However, there is a dissimilarity with the annual cycle in Lhasa, which is affected by southwest monsoon air mass (*Wang, 2014*). This demonstrates that the moisture transported to Central Asia was derived predominantly from the westerlies and the polar air masses. Moreover, the high precipitation d-excess values showed that recycled moisture derived from local sources makes a significant contribution to precipitation, especially in Central Asia. *Yao et al. (2020)* also suggested that the warming and increased moisture content of the atmosphere contributed to the local moisture cycle and increased precipitation recycling in eastern Central Asia.

## CONCLUSIONS

We established the CAMWL and analyzed the spatial–temporal characteristics of precipitation $\delta^{18}O$ and its relationship with meteorological variables and geographical factors based on the precipitation $\delta^{18}O$ values observed at 47 stations over Central Asia, with information from GNIP, TNIP, CHNIP, and TSNIP databases and reference studies.

In addition, we revealed the relationship between precipitation $\delta^{18}O$ and large-scale atmospheric circulation. We made the following conclusions:

1. The CAMWL was established as $\delta^2H = 7.30\delta^{18}O + 3.12$ ($R^2 = 0.95$, $p < 0.01$) for 727 groups of monthly precipitation $\delta^{18}O$ over Central Asia. The precipitation $\delta^{18}O$ over Central Asia ranged from +2‰ to −25.4‰ with a mean of −8.7‰.

2. The precipitation $\delta^{18}O$ over Central Asia was related to meteorological factors. It had a significant positive correlation with temperature, with $\delta^{18}O$-temperature gradients ranging from 0.28‰/°C to 0.68‰/°C. However, the dependence of $\delta^{18}O$ on precipitation was unclear, with a significant effect on precipitation observed only at the Zhangye and Teheran stations, showing $\delta^{18}O$-precipitation gradients of 0.20‰/mm and −0.08‰/mm, respectively.

3. In summer, the latitude was significantly correlated with precipitation $\delta^{18}O$ ($R = 0.54$, $p < 0.05$), while in winter, both latitude and longitude were significantly correlated. The gradient of $\delta^{18}O$/LAT and $\delta^{18}O$/ALT were −0.42‰/° and −0.001‰/m, respectively.

4. The precipitation $\delta^{18}O$ at the Zhangye station showed a significantly positive correlation with the EASMI, and a negative correlation with the IMI at the Urumqi and Teheran stations. In addition, there was a weakly positive correlation between $\delta^{18}O$ and the WCI. Our results suggest that the moisture in Central Asia is mainly transported by westerly circulation and is indirectly affected by the Indian monsoon. Furthermore, the East Asian monsoon can affect the precipitation $\delta^{18}O$ in westerly and monsoon transition regions.

### Funding
This work was funded by the Natural Science Foundation of the Xinjiang Uygur Autonomous Region (2018D01B06). The funders had no role in study design, data collection and analysis, decision to publish, or preparation of the manuscript.

### Grant Disclosures
The following grant information was disclosed by the authors:
Natural Science Foundation of the Xinjiang Uygur Autonomous Region: 2018D01B06.

### Competing Interests
The authors declare that they have no competing interests.

### Author Contributions
- Junqiang Yao conceived and designed the experiments, performed the experiments, authored or reviewed drafts of the paper, and approved the final draft.
- Xinchun Liu analyzed the data, authored or reviewed drafts of the paper, and approved the final draft.

- Wenfeng Hu performed the experiments, analyzed the data, prepared figures and/or tables, and approved the final draft.

## Data Availability

Monthly EASMI and IMI were obtained from Dr. Jianping Li's webpage (http://ljp.gcess.cn/dct/page/65610). Monthly precipitation $\delta^{18}$O data were provided by the GNIP (http://www-naweb.iaea.org/napc/ih/IHS_resources_gnip.html).

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
