# Peer review of "Stable isotope compositions of precipitation over Central Asia"

_PeerJ, doi:10.7717/peerj.11312_

## Round 0.1 · original submission · Major Revisions

I'd like to emphasize that I consider that the reviewers are requesting some very valid clarifications/changes, especially Reviewer 1. You should carefully address the issues in your revised manuscript and in the point-by-point replies to each reviewer. Your revised manuscript will be sent again to reviewers.

·

Basic reporting

The authors present the analysis of stable isotope data in precipitation over central Asia. They used both δ2H and δ18O for construction of meteoric water line, although on couple of sites in the manuscript they explicitly stated they used only δ18O. I suppose the data refer to monthly precipitation samples, since some data were taken from GNIP database. For relation with various climatological and geographical parameters they used only δ18O values, which is understandable. However, since they have both δ2H and δ18O values, I suggest to discuss also deuterium excess that can also help in studying air mass circulation, altitude effect, seasonal changes, etc. Finally, they develop a model, although it is not an ab initio model, but rather an empirical relation between several parameters, to predict/calculate the δ18O in precipitation at Urumqi station. I miss some more detailed discussion on comparison between the measured and calculated values, why there are some large deviations, etc., and on a possible application for paleoclimatology, which was mentioned in the very first sentence of Introduction (although the sentence is rather disordered).
The presented results are interesting for the air mass circulation and isotopes-in-precipitation studies on regional scale, but can be important also for other regions. It shows the importance of GNIP database and other local/regional databases for general knowledge on air mass circulation. This is especially important if we think about the recent climate changes that can already be observed in both different seasonal distribution of precipitation and in changes in their isotopic composition. Therefore, the study may be accepted from the topical point of view, however, the manuscript itself contains various deficiencies that should and must be improved before final accepting for publication. I will list my observations and suggestion below separately as General and Specific comments.

I did not notice the raw data:

Experimental design

1. I found the title, the abstract and the content of the whole paper rather confusion. The title implies the analysis of stable isotopes (usually it includes δ2H and δ18O values) in precipitation, the abstract talks in the beginning of δ18O only (as well as the first line of Introduction), but the a few lines later the meteoric water line is established, and for MWL one needs δ2H data also! Similarly, the first section in Results and discussion is about MWL. The authors should better explain which data they used.
2. The discussion on various dependences of stable isotope composition could be restricted to discussion on δ18O data, because δ2H is well correlated with δ18O through MWL, as the authors also showed. When they produced MWLs by using δ2H data, it could be informative to use also deuterium excess data.
3. Various statistical methods are used in the manuscript, but some essential data are missing in most of the results, such as uncertainties, number of data, R values, p values for significance of correlations. In addition, care should be taken on the number of significant digits. For example, uncertainties in slope and intercept of the MWL line (line 21 in abstract) are given with 4 decimal digits and without uncertainties. From my experience, if uncertainties were correctly taken into account, no more than 2 decimal places should be presented.
4. Please use δ2H instead of D.
5. Central Asia is a wide region, and it would be nice to have the countries in the region by their names, add some political map with countries and the main cities in addition to the map of sampling locations (Figure 1), or at list the names of the main cities on map in Figure 1.
6. I am not a native English speaker, however, I have a long experience with scientific paper written in English. I would suggest language check/proof reading.
7. Line 35 – Could you please explain more about what do you mean by paleoclimatological applications? I have not seen any comment further on.
8. I suggest numbering of all equations.

Validity of the findings

The presented results sound scientifically justified, but some more detailed discussion is missing (see suggestions)

Additional comments

Figures should be improved. Figure 8 is not mentioned in text.
See the specific comments in the attached file.

Reviewer 2 ·

Basic reporting

1.It is suggested to polish the use of grammar and sentences

Experimental design

The paper has a clear structure and correct logic, which is of great significance to the study of precipitation δ18O and water vapor transport in central Asia, even in arid inland regions

Validity of the findings

no comment

Additional comments

1.The research methods and materials section of this paper mainly introduces the sources of data. It is suggested to supplement some detailed introduction of research methods;
2.How accurate is the linear relationship model between precipitation and geographical factors constructed in this paper;
3.It is suggested to add discussion section and make comparative analysis with other relevant studies to explain the consistency and difference; It is better to explain the principle and significance of the conclusion of this study

---

## Round 0.2 · Minor Revisions

While the reviewers acknowledges that your revised version is much improved, there is a need to revise once more your manuscript to address these final suggestions/comments. The reviewer's specific suggestions are included in the annotated manuscript. Please work on this revision and return your manuscript as soon as possible.

Best regards.

·

Basic reporting

Basically, the revised version is much better than the original submission.
References are OK, but should be listed alphabetically.
Minor comments on figures, otherwise OK

Experimental design

no comment

Validity of the findings

only some minor comments given in the annotated pdf file in attachment

Additional comments

The authors took comments of the reviewers into account and the revised version looks better. However, there are some minor comments (for consistency, figure improvements - see the attached file) that should be answered and resolved. For example, you use R, R-squared and CC for correlation coefficient, and I have not seen any explanation what the three different symbols are used.
D has not been replaced with 2H at all places.
Not all references in the text are in italic blue font.
References in the list should be sorted alphabetically
Lettering in Figure 3 should be larger

---

## Round 0.3 · accepted · Accept

Thank you for addressing all the final comments, I consider that your manuscript is now acceptable.